# Cardiac Autonomic Dysfunction and Increased Oxidative Stress in Conventional Cigarettes and E-Cigarettes: Heart Rate Variability as a Cardiovascular Predictor

**DOI:** 10.3390/antiox14121516

**Published:** 2025-12-18

**Authors:** Fernando Sabath de Oliveira Bernardes, Eloisa Maria Gatti Regueiro, Reinaldo Bulgarelli Bestetti, Samuel de Sousa Pereira Araujo, João Paulo Jacob Sabino, Marina de Toledo Durand

**Affiliations:** 1Medical School, University of Ribeirao Preto, Ribeirao Preto 14096-900, SP, Brazil; fernando.sbernardes@sou.unaerp.edu.br (F.S.d.O.B.); eregueiro@unaerp.br (E.M.G.R.); rbestetti@unaerp.br (R.B.B.); 2Physical Therapy School, Barao de Mauá University Center, Ribeirao Preto 14090-062, SP, Brazil; 3Postgraduate Program in Pharmaceutical Sciences, Department of Biophysics and Physiology, Federal University of Piaui, Teresina 64049-550, PI, Brazil; samuel.00.sousa@ufpi.edu.br (S.d.S.P.A.); jacobsabino@ufpi.edu.br (J.P.J.S.)

**Keywords:** tobacco smoking, vaping, oxidative stress, autonomic nervous system, heart rate variability, cardiovascular risk

## Abstract

Conventional and electronic cigarette (e-cig) users face an increased risk of cardiorespiratory diseases, driven by well-characterized pathways involving inflammation and oxidative stress (OS). Conventional cigarettes contain numerous harmful chemicals, such as nicotine and non-nicotine compounds, which produce reactive oxygen species. Although initially considered a safer alternative, the e-cig still generates toxic aldehydes that are capable of triggering oxidative responses. Heart rate variability (HRV) is an important tool for assessing autonomic function and predicting prognosis. Cardiac autonomic dysfunction, indicated by reduced HRV, has emerged as a critical cardiovascular risk factor associated with several diseases. Clinical and experimental studies show that increased OS is directly associated with heightened sympathetic activity and inversely with parasympathetic modulation. This review demonstrates that exposure to conventional cigarettes smoking and e-cigs adversely affects cardiac autonomic function, detectable by a global reduction in HRV that reflects a shift toward sympathetic dominance and a consequent increase in cardiovascular risk. These changes are associated with increased OS due to nicotine and non-nicotine compounds maintaining sympathovagal imbalance in smokers. Thus, we suggest that autonomic dysfunction, detected by HRV, correlates with oxidative responses and may be used as a modifiable risk factor in longitudinal studies involving both smoking modalities.

## 1. Introduction

Over the last few decades, the nicotine-based product industry, such as conventional cigarettes, and electronic nicotine delivery systems (ENDSs), including electronic cigarettes (e-cigs), “vapes” or “pods”, have undergone a significant transformation to increase their appeal and to meet new consumer demands [1]. This phenomenon has boosted the popularity of these products, particularly among teenagers and young adults, due to targeted advertising, marketing, various flavours, nicotine compositions, and a perception of reduced harm [2,3,4,5,6,7,8].

Regardless of the consumption method, smoking is associated with various physiological disorders and represents one of the leading causes of mortality and morbidity worldwide [9], becoming a significant public health issue [2]. ENDSs appeared in the 1970s initially as a less health-damaging option [9]. Nevertheless, in the last two decades with the effective ENDS introduction in the market, research has raised concerns about the real impacts of this new consumer demand and its possible cardiorespiratory, autonomic, and metabolic repercussions [6,7,8,9,10]. Although the ENDSs do not need tobacco combustion like conventional cigarettes, these devices can also deliver toxic chemical substances such as nicotine as well as reactive oxygen species (ROS), which are involved in autonomic dysfunction.

In this context, heart rate variability (HRV) has been demonstrated as a non-invasive tool for assessing alterations in autonomic balance in animals and humans [11,12,13,14,15,16]. HRV has been widely employed in different clinical conditions, including hypertension [17], chronic obstructive pulmonary disease (COPD) [18], and increases in oxidative stress (OS) [17,18,19,20,21,22,23]. Accordingly, the present study aims to review the literature about the impacts of conventional cigarettes and ENDS consumption on cardiac autonomic function, emphasizing the role of HRV as a sensitive tool to detect autonomic alterations associated with high levels of ROS and OS.

## 2. Impact of Conventional and Electronic Cigarettes on Cardiorespiratory Systems

Tobacco smoking remains a leading modifiable risk factor for cardiovascular and respiratory diseases worldwide. Particularly via tobacco combustion, conventional cigarette smoking (CCS) delivers thousands of toxic chemical substances, such as tar and nicotine, and continues to account for millions of deaths each year. Findings confirm the robust associations between tobacco exposure and cardiovascular and respiratory morbidity due to increased risk of coronary artery disease (CAD), myocardial infarction (MI), COPD, lung cancer, and other outcomes [24].

A causal relationship between genetic liability to smoking and the onset of multiple cardiovascular conditions was demonstrated by Larsson and Burgess et al. [25] in a comprehensive Mendelian randomization meta-analysis. While the odds ratio (OR) for CAD in their findings was variable across models, some analyses suggested ORs around 1.33 per unit increase in the lifetime smoking index. Similarly, recent analyses on atrial fibrillation indicate increased risk among smokers, with an OR of 1.23 (95% CI: 1.10–1.36) in specific cohorts such as Finnish populations [24]. Additionally, Al-Halabi et al. [26] demonstrated that smoking 16 to 25 cigarettes per day nearly triples the risk of acute MI. These data collectively reinforce a dose-dependent increase in cardiovascular risk due to cigarette use.

In the respiratory system, tobacco smoking is the leading contributor to COPD, typified by persistent airway inflammation, remodelling, and alveolar destruction [27]. Bircan et al. [28] and Minervini et al. [29] presented associations between smoking and COPD and other several pulmonary conditions such as asthma and COPD–asthma overlap and lung cancer. These findings align with prior meta-analytic studies reporting a lung cancer relative risk of 10.92 (95% CI: 8.28–14.40), a 4.01 risk ratio for COPD, and an OR of 1.61 (95% CI: 1.07–2.42) for adult-onset asthma among smokers [30,31].

A growing body of mechanistic evidence implicates several inflammatory biomarkers in tobacco-related damage. Caspase-1 activation, increased expression of toll-like receptor 4 (TLR4), and elevation of interleukin-6 receptor alpha (IL-6R-α) have been documented in monocytes of smokers [32,33]. In addition, ROS generated by smoking induce epithelial cell apoptosis and activate alveolar macrophages and neutrophils, sustaining a pro-inflammatory pulmonary milieu [27]. These pathways are associated with inflammasome activation, epithelial injury, and systemic immune dysregulation. Such molecular disruptions underpin both respiratory and vascular damage, contributing to chronic conditions such as emphysema, atherosclerosis, and heart failure.

Although initially promoted as a less health-damaging option, ENDSs are increasingly recognized as harmful, particularly to the respiratory system. Exposure to e-cig aerosols similarly results in epithelial damage, airway inflammation, increased ROS production, and endothelial dysfunction, mostly associated with high-powered devices and flavoured liquids [3]. A recent meta-analysis by Shabil et al. [34] quantified the association of e-cig use and risk of COPD: current e-cig users had an OR of 1.48 (95% CI: 1.36–1.61) for COPD compared with nonusers, and ever users had an OR of 1.79 (95% CI: 1.42–2.25). Park et al. [35] and Auschwitz et al. [36] showed that e-cig exposure disrupts epithelial integrity and immune responses in pulmonary tissue, although long-term clinical outcome data remain limited.

Recent studies indicate that e-cig use exerts measurable and deleterious effects on both the cardiovascular and respiratory systems, warranting careful scientific scrutiny and public health attention. E-cig use increases arterial pressure (AP), heart rate (HR), and arterial stiffness while reducing airflow in the lungs [37]. A large body of epidemiological evidence has also indicated that e-cig smokers have a higher risk of developing cardiovascular disease, such as cardiac arrhythmia, hypertension, acute coronary syndromes, and heart failure. In fact, ENDSs are considered an independent risk factor for increased rates of cardiovascular disease occurrence and death [38]. By analyzing the National Health Interview Survey data, Vindhyal et al. [39] found e-cig users had a 50% higher risk of developing MI (OR, 1.56; 95% CI, 1.45–1.68), 30% higher risk of developing stroke (OR, 1.30; 95% CI, 1.20–1.40), and 40% higher risk of circulatory problems (OR—1.43, 95% CI, 1.25–1.65) when compared with nonusers.

Young e-cig users show decreased lung functional capacity and cardiovascular reserve compared with non-smokers, with spirometry indicating an increased risk of respiratory obstruction [40]. Systematic reviews reveal respiratory effects including cough, phlegm, asthma symptoms, and cases of chronic bronchitis and COPD, alongside cardiovascular effects [41]. Additionally, studies indicate that e-cig aerosols also impair endothelial function by activating Nicotinamide Adenine Dinucleotide Phosphate (NADPH) oxidase and reducing flow-mediated dilation (FMD) [42]. Elevated biomarkers such as soluble intercellular adhesion molecule-1 (sICAM-1) and high-sensitivity C-reactive protein (hs-CRP) have been detected in dual users of tobacco and e-cigs, signalling vascular inflammation and endothelial compromise [43].

The mechanisms underlying cardiovascular damage include hyperlipidaemia, sympathetic dominance, endothelial dysfunction, DNA damage, and macrophage activation, with OS and inflammation serving as unifying pathways [44]. Animal studies demonstrate that long-term exposure leads to lung architectural changes, immune gene dysregulation, and low-grade inflammation [37]. These findings indicate substantial cardiorespiratory health risks associated with e-cig use. The widespread use of e-cigs has the potential to perpetuate a new epidemic of chronic cardiorespiratory disease.

Indeed, a conventional cigarette or ENDS significantly increases the risk of cardiopulmonary disease by means of well-characterized inflammation and OS [45]. In a murine model, Mancuso et al. [46] reported that e-cig exposure exacerbated cerebral ischemic injury and elicited OS comparable to that of combustible cigarettes. Additionally, both CCS and e-cig aerosols compromise mitochondrial bioenergetics and disrupt autophagic flux, thereby promoting structural and functional pulmonary decline [47].

The oxidative process also plays an important role in the development of endothelial dysfunction, a precursor to atherosclerosis. Combustible CCS impairs nitric oxide (NO) bioavailability via endothelial Nitric Oxide Synthase (eNOS) uncoupling and heightened peroxynitrite production, contributing to vascular stiffness, hypertension, CAD, and stroke [48]. Endothelial dysfunction, frequently assessed via FMD, is consistently impaired in smokers, though partial reversibility is observed following cessation [2,48,49].

## 3. Effect of Conventional and Electronic Cigarettes on Oxidative Stress

Oxidative stress refers to an imbalance between the production of ROS and the body’s intrinsic antioxidant defences, resulting in molecular damage to cellular components such as DNA, proteins, and lipids [27]. Tobacco use is a well-established exogenous inducer of OS and it has long been implicated in the pathogenesis of chronic diseases due to its pro-oxidant properties. Even though ENDSs were initially promoted as a less harmful alternative, growing scientific evidence demonstrates that e-cigs can equally disrupt redox homeostasis and elicit oxidative damage [27,49]. Thus, both combustible tobacco products and ENDSs contribute significantly to OS.

CCS generated by tobacco combustion contains over 7000 chemical constituents, many of which possess the capacity to generate ROS. A single puff of a cigarette emits approximately 10^15^ free radicals in both gaseous and tar phases, including hydroxyl radicals (^•^OH), superoxide anions (O_2_^•−^), hydrogen peroxide (H_2_O_2_), and peroxynitrite (ONOO^−^). These species induce direct molecular damage and activate redox-sensitive signalling cascades such as Nuclear Factor kappa-light-chain-enhancer of activated B cells (NF-κB) and Mitogen-Activated Protein Kinase (MAPK), which facilitate inflammation and apoptotic cell death (Figure 1) [27].

Tobacco smoking has been directly associated with elevated levels of OS biomarkers, including soluble NOX2-derived peptide (sNOX2-dp), F2-isoprostanes (F2-IsoP), malondialdehyde (MDA), and 8-hydroxy-2′-deoxyguanosine (8-OHdG), hallmarks of lipid peroxidation and oxidative DNA injury [21,49]. These ROS-mediated effects impair mitochondrial dynamics, decrease glutathione availability, and compromise cellular energy metabolism. Furthermore, prolonged exposure to CCS enhances the systemic expression of pro-inflammatory cytokines such as Interleukin-6 (IL-6), Tumour Necrosis Factor alpha (TNF-α), and Interleukin-1beta (IL-1β), further amplifying ROS generation and tissue injury (Figure 1) [43]. Endogenous antioxidant pathways such as the Nrf2/ARE response can also be activated by CCS, but the protective activation is often insufficient to counterbalance ROS accumulation [50].

ENDSs, by contrast, generate aerosols by means of the heating of e-liquids composed primarily of nicotine, propylene glycol, vegetable glycerine, and various flavouring agents. Although these devices do not involve combustion, thermal decomposition of their constituents produces toxic aldehydes, including formaldehyde, acrolein, and acetaldehyde, capable of triggering significant oxidative responses [42]. Some studies demonstrated that although e-cigs led to a significant degree of alterations in OS, they seemed to have less pronounced adverse effects as compared with CCS [51]. Moheimani et al. [52], Shahab et al. [53], and Christensen et al. [43] showed that e-cigs are responsible for elevated levels of F2-IsoP and low-density lipoprotein oxidizability (oxLDL), biomarkers of OS and inflammation, in comparison to non-smokers (Figure 1) [42]. Conversely, Taylor et al. [50] observed that exposure to e-cig aerosol extracts did not result in a significant rise in intracellular ROS or depletion of glutathione in bronchial epithelial cells, unlike CCS, which induced an increase of 83% in ROS production and a reduction of 98.6% in the GSH:GSSG ratio. Nonetheless, subsequent studies confirm that chronic exposure to e-cig vapour can still induce oxidative DNA damage, mitochondrial dysfunction, and overexpression of pro-inflammatory genes [47].

The impact of conventional cigarettes or ENDSs on the human body extends beyond the oxidative and inflammatory effects already outlined, also interfering with cardiac autonomic regulation. Continuous exposure to CCS or ENDS aerosols induces physiological changes that disturb the balance between the sympathetic and parasympathetic systems. This imbalance may be reflected in changes in HRV, a sensitive marker of autonomic influence on cardiac function.

## 4. Cardiac Autonomic Regulation and Heart Rate Variability

The heart–brain axis works by means of a mechanism in which the autonomic nervous system (ANS) establishes an equilibrium between them and their function [54]. The ANS controls cardiac activity, increasing or decreasing it by means of sympathetic and parasympathetic stimulation, respectively. What modulates these signals depends on the trigger, which can be a fight-flight response, diseases, lifestyle, and many others. Since the heart is influenced by the brain, and vice versa, the activity of the ANS over the heart has become one of the sources of study to understand the influences from these stimuli and their consequences [11,55,56].

HRV is defined as method for studying instantaneous variations in HR or, more precisely, for evaluating the time-related oscillations in the interval of two consecutive heartbeats, using the R-R intervals (iRRs) [11]. Accordingly, HRV analysis is a reliable and non-invasive procedure for assessing the coupling between the autonomic nervous system (ANS) and the sinoatrial node [11,54]. Using a 24 h electrocardiogram (ECG) record, it is possible to extract successive iRRs and analyse HR over time, providing correlation with different situations in response to the ANS. This analysis has emerged as an indicator of autonomic balance and offers insights into healthy lifestyle or pathological states (Figure 2) [11,54,55,56,57,58].

Cardiac contraction activity occurs in a precise way, as it is a watch, primarily because of a “natural” pacemaker, the sinus node, which is responsible for controlling the heart rate [57]. In the absence of autonomic innervation, cardiac contractile activity proceeds with remarkable precision, akin to the regularity of a metronome, exhibiting consistent intervals between successive heartbeats. This phenomenon is primarily attributable to the sinoatrial node, the intrinsic pacemaker. Nevertheless, both the sympathetic and parasympathetic divisions influence the sinoatrial node function, each playing a significant role in modulating the heartbeat intervals [57].

Sympathetic innervation is responsible for increasing the activity of the sinoatrial node by means of epinephrine and norepinephrine release and β1-adrenergic receptor activation. This leads to an increase in intracellular cyclic AMP (cAMP) levels which accelerates the slow diastolic depolarization (phase 4) of the slow action potential and increase its amplitude. The underlying intracellular mechanism is an increase in inward sodium (If) and calcium currents (ICa^2+^). Conversely, the parasympathetic system, via the vagus nerve, releases acetylcholine and activates the M2-muscarinic receptor, reducing cAMP and opening acetylcholine-dependent potassium channels, increasing outward potassium current (IKACh). This leads to greater hyperpolarization, slowing phase 4 of the action potential and reducing the depolarization rate of the sinoatrial node [11,57].

It should be pointed out that these intracellular mechanisms have significant implications for HRV. While the parasympathetic modulation of the HR occurs at higher frequencies due to rapid intracellular process that rely on potassium channels, the sympathetic activity operates within lower frequency intervals as a result of mechanisms dependent on cAMP concentration [11,57]. On that account, different approaches for measuring the oscillations of the HR are described in the literature.

Researchers use two primary methods to analyze HRV: the linear method, which includes the evaluation of the time and frequency domains, and the nonlinear method, based on the Poincaré plot and acceleration or deceleration capacity [11,55,56,57,58]. The linear method is the simplest and most used in the literature, offering a general assessment of heart autonomic modulation, as described below.

-Time Domain—this is the most common and the simplest way to evaluate HRV. This method seeks to understand how the HR varies in a period of time. By analyzing the iRR on the ECG, it is possible to determine what is called the Normal-to-Normal (NN) interval or the instantaneous HR, calculate the mean HR or the mean NN interval, and even compare NN intervals on the same ECG. Additionally, some parameters can be investigated using the NN intervals, such as the standard deviation of these intervals (SDNN), the standard deviations of the means (SDANNs), the root mean square of the successive differences between normal iRRs (RMSSD), and the percentage of successive NN intervals greater than x ms (pNNx), of which the most commonly used threshold in humans is 50 ms (pNN50). The last two reflect the influence of parasympathetic autonomic modulation on the heart [11,55,56,57,58].-Frequency Domain—unlike the time domain, data are analyzed based on the HR frequency variation and its distribution, represented by power spectral density. This method applies mathematical algorithms, e.g., the Fast Fourier Transformation or autoregressive modelling, which can assist in visualizing the HRV density distribution (Figure 2). It can correlate more closely with the sympathetic and parasympathetic influence on the heart. There are three basic spectral components of this domain and a ratio between them that can be calculated: Very Low Frequency (VLF, 0.0033–0.0400 Hz), Low Frequency (LF, 0.04–0.15 Hz), High Frequency (HF, 0.16–0.40 Hz), and LF/HF ratio. The first two correlate more with sympathetic activity, the third with parasympathetic modulation, and the last one is used as a parameter to assess the balance between sympathetic and parasympathetic activities [11,55,56,57,58].

## 5. Heart Rate Variability as a Health Predictor

Over the years, HRV has been widely used as a tool in research and clinical practice as it is an easily accessible and non-invasive method. Some factors such as age, physiological and pathological states, and lifestyle, genetic, and environmental factors can affect HRV and serve as a starting point to study how they can influence cardiac autonomic function, as well as to assess cardiovascular risk [56,58,59,60].

Each situation can manifest different HRV. After an acute MI, a reduction in HRV is expected, indicating diminished vagal activity and sympathetic predominance to the heart. These marks are associated with increased mortality, arrhythmic events, and a poor prognosis for those patients [11,55]. Likewise, hypertensive patients present an important reduction in HRV a high LF component, indicating cardiac sympathetic overactivity and reduced parasympathetic modulation. These autonomic changes induce significant structural and functional alterations in the heart, which are also associated with higher morbidity and mortality outcomes [11,55].

Diabetes can also induce severe autonomic imbalance due to diabetic neuropathy, which affects both sympathetic and parasympathetic fibres. As a result, HRV reduces due to vagal derangement, reflecting a dysautonomia, which can be associated with a higher risk for cardiovascular events, such as sudden cardiac death [55,56]. Additionally, the consumption of substances such as drugs, alcohol, and tobacco leads to an autonomic imbalance, characterized by increased sympathetic and reduced parasympathetic activity, ultimately lowering HRV [55,56]. Conversely, maintaining a healthy lifestyle increases HRV, which is associated with better overall health status (Figure 2) [16,23,55,56,58].

Physical activity has been used as a key component of treatment for several diseases due to its role in cardiac autonomic regulation [16,61]. Exercise programmes stimulate an increase in HRV, suggesting higher cardiac parasympathetic modulation and/or reduced sympathetic activity, associated with better metabolic, inflammatory, and OS parameters [61,62]. Despite the different workout programmes, moderate to highly intensive exercises demonstrated higher parasympathetic and HRV indices, indicating improved cardiac autonomic function [61,62,63].

HRV can also be used in other scenarios. Some studies use this method to evaluate HRV in small-for-gestational-age newborns [59], demonstrating a higher sympathetic activity in these infants compared with those appropriately sized for gestational age. In other cases, the HRV is also used to assess how a person’s emotional state affects the ANS. Some psychological conditions, including anxiety and mood disorders, show a global reduction in HRV, indicating a shift toward increased sympathetic relative to parasympathetic activity [56,60].

HRV can be used not only in humans, but also in animals; the study of HRV serves as a tool to diagnose arrhythmia in horses [64], to evaluate dysautonomia due to chagas diseases in domestic canines [65], or to observe the effects of substances on the HRV in rats [13] and in mice models [14,15,66]. Recently, Kucera et al. [67] demonstrated that mice exposed to e-cig aerosols from 5% nicotine salt showed a decrease in SDNN and RMSSD during the washout period and an increase in cardiac arrhythmias, suggesting a shift to sympathetic dominance activity and higher cardiovascular risk.

Overall, HRV stands out as an important non-invasive biomarker to assess autonomic function, being used as risk stratification and prognostic tool [68,69]. In a prospective cohort study, patients with cardiac heart failure were followed up for major adverse cardiovascular events. Those individuals who experienced these events presented reduced time domain values and lower HRV during analyses [69]. In another retrospective cohort study, analysis of the HRV parameters were also used as a predictor of mortality and admission to intensive care unit due COVID-19. In these cases, patients with higher HRV presented significantly less mortality risk and fewer admissions [70]. In patients with COPD and/or Obstructive Sleep Apnea, HRV analysis was also considered as an important biomarker to evaluate the prognosis and response to treatment, as these diseases have a significant impact on the cardiovascular system [71].

## 6. The Role of Oxidative Stress in Cardiac Autonomic Dysfunction

Cardiac autonomic dysfunction, detected by HRV, has been associated with OS in a variety of clinical conditions. Experimental studies have shown that ROS can stimulate sympathetic nervous system activity in the brain and in the peripheral ANS. Campese et al. [72] postulated that such an action could be the consequence of decreased NO and IL-1β cytokine activity.

Indeed, there is evidence demonstrating that ROS are produced in key central regions associated with cardiovascular function, such as the hypothalamus, nucleus of the solitary tract, and rostral ventrolateral region of the medulla. These regions play a crucial role in determining basal sympathetic activity [73,74,75,76,77,78,79]. Additionally, locally produced ROS in the baroreceptor region or central nervous system directly affect the baroreceptor reflex and contribute to dysautonomia [80,81]. Li et al. [80] found that ROS generated in the carotid bulb of atherosclerotic rabbits reduced the carotid sinus nerve response to elevated AP. This reduction was reproduced by exogenous ROS but prevented by their inhibition (Figure 1).

In clinical scenarios, OS can also be associated with cardiac autonomic dysfunction in patients with different chronic diseases (Table 1). Fadaee et al. [19] observed that 16 of 78 (20%) patients with chronic kidney disease (CKD) had elevated levels of F2-IsoP, which was associated with cardiac autonomic dysfunction disorder. They correlated plasma F2-IsoP with the HRV indices in both time and frequency domain and found an inverse correlation with SDNN, VLF, LF, and total power parameters. Moreover, they also detected that plasma F2-IsoP independently predicted SDNN, a global measure of HRV, indicating that CKD patients with elevated OS display evidence of autonomic dysfunction.

The association between plasma indices of OS and cardiac autonomic dysfunction has been studied in patients with prehypertension. Thyagarajan et al. [20] found that the HRV parameters in both the time and frequency domain concerning the parasympathetic system were decreased in prehypertensive young adults in comparison with control subjects. In addition, the authors showed that plasma OS parameters (TBARS and total antioxidant capacity) were deranged in prehypertensive patients. Finally, they discovered that the RMSSD index, a marker of parasympathetic derangement, was correlated with plasma indices of OS. The study, therefore, suggested that prehypertensive patients have an association between increased OS and abnormal cardiovagal modulation.

A similar condition has been observed in patients with systemic arterial hypertension in comparison with normotensive ones. Pavithran et al. [17] examined patients newly diagnosed with systemic arterial hypertension and normotensive controls. These patients underwent HRV studies in both time and frequency domain, which were correlated with plasma measurement of OS and total antioxidant status. The authors observed that several indices of parasympathetic cardiac modulation (SDNN, RMSSD, HFabs, and HFnu) inversely correlated with MDA in hypertensive patients but not in normotensive. Furthermore, the authors also noted that MDA correlated positively with an index of sympathetic activity (LFnu) in hypertensive patients but not in those with normal systemic AP. Collectively, such data shows an inverse correlation between plasma parameters of OS and parasympathetic activity and a positive correlation between OS and sympathetic activity in patients with systemic arterial hypertension.

Recent findings by Promsrisuk et al. [18] also suggest that OS plays a potential role in impairing cardiac autonomic control in COPD patients. These authors investigated a link between MDA levels, an OS marker, pulmonary function, and parameters of HRV in clinically stable COPD patients and healthy male controls. COPD patients exhibited reduced time domain indices (SDNN and RMSSD) and HF component and increased LF component and LF/HF ratio in supine and head-up tilt positions compared with controls. In addition, MDA levels were 2.3 times greater in COPD patients and they were positively correlated with LF and LF/HF ratio but inversely correlated with SDNN and pulmonary function parameters in both positions. These findings indicated that OS may induce sympathetic overactivity in patients with COPD.

The situation seems to be somewhat different in patients with diabetes mellitus with peripheral autonomic disorder and superimposed cardiac autonomic dysfunction detected by HRV. Ziegler et al. [21] studied 22 diabetes mellitus patients with both conditions in comparison with patients without one or both and healthy subjects. They observed that the plasma parameters of OS (8-iso-PGF2α, O_2_^•−^, and ONOO^−^) in such patients were higher compared with control subjects but were similar to those in patients with peripheral neuropathy without cardiac autonomic dysfunction. This suggests that OS is enhanced in diabetic patients with polyneuropathy, without further significant increase in relation to superimposed autonomic neuropathy.

In fact, the rise in plasma glucose likely drives increased OS parameters, which may induce cardiac autonomic dysfunction. Thiyagarajan et al. [82] found that in prediabetic subjects, autonomic imbalance, hyperinsulinemia, insulin resistance, OS, inflammatory response, and impaired fasting glucose interact continuously—each factor reinforcing the others and contributing to disease progression. Additionally, Ziegler et al. [83] found that about one year after type 2 diabetes was diagnosed, subepidermal SOD2 expression increased by approximately 60%, likely as a compensatory response to ROS formation. The findings also reveal a correlation between this increase and longer diabetes duration, a shift towards cardiac sympathetic dominance, and reduced vagal tone, as evidenced by an elevated LF/HF ratio and decreased RMSSD. Recent findings by Al-Saoudi et al. [84] indicate that, in young adults with type 1 diabetes, lipid peroxidation correlates with an increased LF/HF ratio, while glucotoxicity is linked to reduced SDNN values.

Despite the dearth of papers on the association between the HRV indices and plasma parameters of OS, it seems that plasma parameters of OS are associated with indices of sympathetic activity and inversely correlated with those of parasympathetic activity. However, this link between oxidative metabolism and the activity of ANS is still limited and deserves further elucidation. A recent review by Yelisyeyeva et al. [23] also discusses this relationship between autonomic regulation, HRV, and redox homeostasis during ageing.

**Table 1 antioxidants-14-01516-t001:** Key parameters of clinical studies that associated oxidative stress (OS) with autonomic dysfunction assessed by means of heart rate variability.

Reference	Clinical Condition	Size Sample (*n*)	Study Design	OS Parameters	HRV Parameters
Fadaee et al., 2016 [19]	Chronic kidney disease	Normal F2-IsoP *n* = 62 Elevated F2-IsoP *n* = 16	Cross-sectional	⭡ F2-IsoP (*n* = 16) ↔ GPx ↔ TAC	⭣ SDNN ⭣ Total power ⭣ VLF ⭣ LF ⭣ LF/HF ratio
Thyagarajen et al., 2013 [20]	Prehypertension	Prehypertensive *n* = 97 Normotensive *n* = 81	Cross-sectional	⭡ TBARS ⭣ TAC	⭣ SDNN ⭣ RMSSD ⭣ pNN50 ⭣ HFnu ⭡ LFnu ⭡ LF/HF ratio
Pavithran et al., 2008 [17]	Hypertension	Hypertensive *n* = 36 Normotensive *n* = 14	Cross-sectional	⭡ MDA ⭣ TAC	⭣SDNN ⭣ RMSSD ⭣ RR triangular index ⭣ LF ⭣ HF ⭣ Total power
Promiusk et al., 2023 [18]	COPD	COPD *n* = 50 Controls *n* = 50	Cross-sectional	⭡ MDA	⭣ SDNN ⭣ RMSSD ⭣ HFnu ⭡ LFnu ⭡ LF/HF ratio
Ziegler et al., 2004 [21]	Type 1 and 2 diabetes with PN and/or CAN	PN^−^/CAN^−^ *n* = 62 PN^+^/CAN^−^ *n* = 105 PN^+^/CAN^+^ *n* = 22 Control *n* = 85	Cross-sectional	⭡ 8-iso-PGF_2α_, ⭡ O_2_^•−^, ⭡ONOO^−^ ⭣ vitamin E-to-lipid ratio ⭣ vitamin C CAN did not raise OS parameters further.	CAN was defined as three or more abnormalities among the HRV indexes.
Thiyagarajan et al., 2012 [82]	Prediabetes	Prediabetes *n* = 47 Controls *n* = 76	Cross-sectional blinded	⭡ TBARS ⭡ TAC	⭣ SDNN ⭣ RMSSD ⭣ pNN50 ⭣ Total power ⭣ HFnu ⭡ LFnu ⭡ LF/HF
Ziegler et al., 2015 [83]	Recently diagnosed type 2 diabetes	Diabetes *n* = 69 Controls *n* = 51	Cross-sectional	⭡ Subepidermal SOD2	⭣ RMSSD ⭡ LF/HF ratio
Al-Saoudi et al., 2022 [84]	Type 1 diabetes	*n* = 151	Cross-sectional	⭡ CML ⭡ G-H1	⭣ SDNN ⭡ LF/HF ratio

Abbreviations: ↑ increase; ↓ decrease; ↔ no change; CML, Ne-(carboxymethyl)-lysine; CAN, cardiovascular autonomic neuropathy; COPD, chronic obstructive pulmonary disease; F2-IsoP, F2-isoprostanes; G-H1, glyoxal-derived hydroimidazolone 1; GPX, glutathione peroxidase activity; HF, high frequency; HFnu, high-frequency power normalized; LF, low frequency; LFnu, low-frequency power normalized; MDA, malondialdehyde; PN, polyneuropathy;. RMSSD, root mean square of the successive differences between normal iRRs; SDNN, standard deviation of NN intervals; SOD2, superoxide dismutase 2; TAC, total antioxidant capacity; TBARS, thiobarbituric acid reactive substance.

## 7. Cardiac Autonomic Dysfunction in Conventional and Electronic Cigarette Smokers and the Role of Oxidative Stress

A key mechanism linking CS to cardiovascular risk is cardiac autonomic dysfunction, which correlates with smoking intensity [85,86]. Our group demonstrated that rats exposed for 30 days to CCS showed systemic changes and autonomic cardiocirculatory dysfunction, depending on the daily exposure dose. Although basal cardiovascular parameters, spontaneous baroreflex sensitivity, and cardiovascular variabilities were similar among rats exposed to CCS of one or two (CS2) cigarettes/animal/day, CCS exposure progressively blunted the bradycardia response to phenylephrine, increased sympathetic tone and chemoreflex sensitivity, and reduced the intrinsic HR [13].

Other studies have also demonstrated autonomic changes in unanesthetised rats exposed to acute and chronic CCS exposure. Houdi, Dowell, and Diana [87] and Nakamura and Hayashida [88] observed that acute CS exposure reduced the HR and cardiac output and increased the AP, renal sympathetic nerve activity, plasma norepinephrine, and total peripheral resistance in unrestrained conscious rats. On the other hand, Valenti et al. [89] found only mild alterations in autonomic and baroreflex function of rats exposed to CCS for 3 weeks.

In humans, there is consistent evidence that whether acute or chronic, passive or active, smoking alters autonomic function. Data from the literature describes that this disruption is characterized by decreased HRV, parasympathetic modulation and baroreflex sensitivity, and increased sympathetic drive, which increase cardiac morbidity and mortality [90,91,92,93,94,95,96,97]. Of note, there is also evidence in humans demonstrating that the number of cigarettes and the current smoking intensity are crucial for determining the effect of smoking on the ANS [36,38,39,40,41]. Freire et al. [86] demonstrated that the frequency and time of passive CCS exposure has a negative influence on the hemodynamic response, ANS, and pulmonary function in humans. Additionally, subjects who smoke more than 10 cigarettes/day or 20 packs/year showed lower HRV [81,94]. Also, the HRV indexes are reduced by more than 9% every 10 g of smoked tobacco daily [92].

Clinical data have shown that CCS exposure promotes cardiac autonomic dysfunction characterized by sympathetic overactivation. The main mechanism involved in this autonomic imbalance is the direct pharmacological action of nicotine in its receptors located in peripheral autonomic ganglia and the adrenal medulla, increasing catecholamine release [90,92,96,97,98,99,100,101,102,103,104,105,106]. Sympathoexcitatory effects may also result from the influence of nicotine on chemoreflexes [107]. Argacha et al. [108] found that nicotine enhances peripheral chemoreflex sensitivity to hypoxia in non-smokers, which increases sympathetic drive.

Non-nicotine components may also cause sympathoexcitation by upregulating nicotine receptors [101] or inhibiting monoamine oxidase to reduce catecholamine degradation [106]. Additionally, non-nicotine smoke inhalation can activate olfactory afferents which activate both sympathetic and parasympathetic responses, affecting cardiovascular and respiratory functions in conscious rats [88]. Of note, Middlekauff, Park, and Moheimani [90] proposed that non-nicotine components, such as fine particulate matter, associated with the direct effect of nicotine, are responsible for sympathoexcitation in smokers by means of a complex mechanism involving OS, inflammation, a decrease in nitric oxide production, and baroreflex suppression. Additionally, there is evidence that chronic sympathetic overactivation also contributes to oxidative stress [109,110,111,112,113], perpetuating a self-sustaining detrimental cycle between sympathetic nerve activity and an increase in ROS production. Thus, increased OS is a crucial mechanism which contributes to sympathoexcitation [90].

One mechanism which can explain this positive feedback between the sympathetic nervous system and OS is the activation of the “splenocardiac axis” or the “neural–hematopoietic inflammatory axis”. Libby et al. [114] proposed that this axis is an important pathway connecting the brain, autonomic nervous system, and the hemopoietic system (bone marrow and spleen). The authors hypothesized that this axis is likely to be responsible for the increased risk of future acute ischemic cardiovascular events associated with elevated sympathetic tone in humans. Stress settings that trigger sympathetic outflow from the central nervous system, such as acute myocardial infarction, pain, and mental stress, mobilize leukocyte progenitors from their bone marrow niche via β3-adrenergic stimulation. These progenitor cells can migrate to the spleen, where they can multiply in response to hematopoietic growth factors, increasing pro-inflammatory cytokines (IL-1β and TNFα) and the methylation of NFκB and STAT3 genes [114,115]. Elevated inflammation can, in turn, further increase OS, creating a positive feedback loop that sustains damaging effects of both inflammation, OS, and sympathetic overactivation [115,116,117,118] (Figure 1).

Indeed, Boas et al. [119] demonstrated that smokers showed an increase in metabolic activity in both the spleen and blood vessel wall, indicating that sympathetic overactivation can initiate this axis. Additionally, Ruedisueli et al. [120] observed that the sympathetic nervous system, as activated by the amygdala, plays a role in provoking inflammatory monocyte proliferation and instigating atherosclerotic development. On that account, these findings indicate possible mechanisms by which cigarettes may lead to increased risk of future cardiovascular events.

Furthermore, converging evidence indicated that the ANS is largely involved in the regulation of the “inflammatory reflex” [121]. This reflex is activated by an inflammatory response by means of afferent signals via the vagus nerve to the nucleus of tract solitary. A subsequent efferent signal via the parasympathetic system inhibits pro-inflammatory cytokine synthesis, known as the cholinergic anti-inflammatory pathway. Conversely, the sympathetic nervous system, influenced by this pathway, can increase inflammation in the early stages of inflammation [122,123] (Figure 1). In a meta-analysis of human studies, Williams et al. [124] showed that markers of inflammation were negatively associated with some HRV parameters (SDNN and HF). Hence, the authors proposed that the HRV indices can be used to assess the activity of inflammatory processes. Accordingly, we are proposing that HRV can also be used as an index of OS in CS and e-cig smokers, as inflammatory and oxidative process act as cooperative and synergistic partners in the pathogenesis of cardiovascular diseases [116,125].

In fact, experimental data indicates an association between elevated OS in the brain and autonomic dysfunction in rats exposed to CS. Valenti et al. [89] found that three weeks of side stream CS (SSCS) exposure heightened cardiovascular responses to acute administration of the catalase inhibitor 3-Amino-1,2,4-triazole (AZT) into the 4th ventricle of rats. They observed that AZT injection into the 4th V increased the basal HR more in rats exposed to SSCS and reduced their bradycardic peak compared with fresh air-exposed rats. Furthermore, increased OS in the lungs leads to heightened sympathetic activation by means of the stimulation of lung afferent neurons.

As previously mentioned, several studies showed that cigarette use increases ROS production, such as H_2_O_2_, O_2_^•−^, ^•^OH, and ONOO^−^ [27,126]. Nevertheless, data that associate autonomic dysfunction with OS in tobacco smokers are scanty (Table 2). The study of Trikunakornwong and Suwanprasert [126] was the only one which investigated this relationship by means of HRV assessment and NO and sLOX-1 production during acute smoking in smokers compared with non-smokers. At the basal level, they found a positive correlation between the serum soluble lectin-like oxidized low-density lipoprotein receptor-1 (sLOX-1) and AP and reduced HRV in smokers. After 5 min of smoking, the serum NO level decreased and it was negatively correlated with the LF/HF ratio, indicating an overall sympathetic excitability. These data indicated that the overproduction of sLOX-1 and decreased bioavailability of NO may impair autonomic control in chronic smokers after acute single cigarette smoking, which immediately elevated sympathetic activity and blunted vagal tone.

Although some studies showed that e-cigs may present a lower cardiovascular risk compared with traditional cigarettes, they are not safer and can also pose significant cardiovascular risks [127,128]. In a recent systematic review, Yacoub et al. [109] observed that both acute and chronic e-cig exposure alters the HR, AP, and cardiovascular autonomic regulation, heightening risks of arrhythmias and MI. Passive exposure to e-cig emissions may also have an effect on cardiac autonomic function. Lee et al. [129] observed that a short-term (<15 min) second-hand exposure to nicotine from e-cigs reduced the HRV parameters (SDNN and ASDNN) and shortened the corrected QT interval in a small group of healthy non-smokers. Additionally, these authors found that longer second-hand e-cig exposure increases cardiac effects, suggesting nicotine may have both immediate and cumulative impacts on the heart.

As in tobacco smoking, studies demonstrated that nicotine is the main instigator of the cardiovascular adverse effects that accompany ENDSs [130,131,132,133]. Nguyen et al. [130] demonstrated that higher levels of nicotine acutely delivered by the 4th generation e-cig increased the HR and AP and produced changes in the HRV pattern associated with sympathetic overactivity. Nonetheless, there are data indicating that both nicotine and non-nicotine e-cig components can increase cardiac risk [132]. Nagy et al. [132], by means of a systematic review of case reports, identified cardiovascular complications associated with both nicotine and non-nicotine vaping, with half of the cases resulting in cardiac arrest, likely due to arrhythmias from sympathetic predominance. Conversely, in a systematic review including 19 studies, Garcia, Gornbein, and Middlekauff [131] found that acute vaping increased the HR and AP less than CS. Furthermore, their analysis indicated that nicotine, rather than non-nicotine constituents in e-cig aerosol, was responsible for these sympathoexcitatory effects.

Recently, mice-based model studies evaluated the effects of different formulations of e-cig on HRV. Overall, the e-cig reduces the HR and increases HRV during inhalation exposure, with inverse post-exposure effects [133,134,135,136]. Kucera et al. [67] demonstrated that exposure to >2.5% nicotine reduced the HR and augmented the HRV indexes during e-cig exposure, denoting vagal activation. Conversely, after exposure, an increase in the HR and spontaneous ventricular arrhythmias and a decrease in time domain parameters (SDNN and RMSSD) were noted, indicating a shift to sympathetic dominance. Using menthol e-cigs, Ramalingam et al. [135] observed that mice exposed to this formulation showed an increase in the HR and AP, spontaneous ventricular arrhythmias, and in urine epinephrine and a reduction in time domain indices (RMSSD and SDNN). In both studies, pretreatment with atenolol abolished e-cigarette-induced arrhythmias, suggesting the involvement of β1-adrenoceptors in sympathoexcitation. Also, the cardiac autonomic imbalance persisted even after 3 weeks of cessation, as mice exposed to e-cig aerosols showed an altered chronotropic response to acute stress [135].

Evidence that associates autonomic impairment and OS in e-cig smokers is also scarce (Table 2). In humans, Moheimani et al. [128] observed that habitual e-cig users showed a reduced HF and an augmented LF and LF/HF ratio compared with nonuser control participants. These changes indicated a shift in cardiac autonomic balance toward sympathetic predominance and decreased vagal tone, which were also associated with an increase in oxLDL, a measure of OS, in habitual e-cig users. These abnormalities are known mechanisms which increase cardiovascular risk in diverse patient populations with and without known cardiac disease.

In mice, Castellanos et al. [136] showed that 3 weeks of e-cig aerosol exposure altered the HR and HRV within minutes during and after exposure. The authors observed rapid HR reduction and HRV elevation during e-cig events and a decrease in time domain parameters (SDNN, RMSSD, and pNN6) post-exposure. The timing and frequency of e-cig exposure influenced these changes, with marked effects in weeks 2 and 3. Furthermore, e-cig exposure increased hydroxyl radical generation in mouse bronchoalveolar lavage fluid, which can be a mechanism responsible for mediating autonomic dysfunction. As previously mentioned, autonomic shifting influences ROS production, which may also alter HRV and create a feedback loop that maintains ongoing sympathoexcitation.

**Table 2 antioxidants-14-01516-t002:** Key parameters of studies that associated oxidative stress (OS) and autonomic dysfunction, assessed by means of heart rate variability (HRV), with conventional or electronic cigarettes (e-cigs).

Reference	Species	Cigarette Type	Size Sample (*n*)	Study Design	OS Parameters	HRV Parameters
Trikunakornwong and Suwanprasert, 2019 [126]	Human	Conventional cigarette (acute)	Habitual smokers *n* = 30 Non-smokers *n* = 30	Cross-sectional	⭣ NO ⭡ sLOX-1	⭣ SDNN ⭣ RMSSD ⭣ pNN50 ⭣ Total power ⭣ HFnu ⭡ LFnu ⭡ LF/HF ratio
Moheimani et al., 2017 [128]	Human	e-cig (chronic)	Habitual e-cig users *n* = 16 Controls (non-tobacco or e-cig) *n* = 18	Cross-sectional case control	⭡ oxLDL ⭣ PON-1 ↔ HOI	⭣ HFnu ⭡ LFnu ⭡ LF/HF ratio
Moheimani et al., 2017 [52]	Human	e-cig (acute) with or without nicotine	Healthy volunteers *n* = 33	Open-label, randomized, crossover	↔ HOI ↔ LDL-Ox ↔ PON-1	E-cig with nicotine: ⭣ HFnu ⭡ LFnu ⭡ LF/HF ratio
Castellanos et al., 2025 [136]	Mice	e-cig (post exposure)	C57BL/6J *n* = 6	Experimental	⭡ ·OH	⭣ SDNN ⭣ RMSSD ⭣ pNN6 ⭡ HF ⭡ LF ⭡ LF/HF ratio

Abbreviations: ↑ increase; ↓ decrease; ↔ no change; HOI, HDL antioxidant index; HF, high frequency; HFnu, high-frequency power normalized; LF, low frequency; LFnu, low-frequency power normalized; NO, nitric oxide; oxLDL, LDL oxidizability; PON-1, paraoxonase-1 activity; RMSSD, root mean square of the successive differences between normal iRRs; SDNN, standard deviation of NN intervals; sLOX-1, soluble lectin-like oxidized low-density lipoprotein receptor-1.

## 8. Conclusions

In conclusion, this review indicates that both conventional cigarettes and ENDSs induce cardiac autonomic dysfunction characterized by a shift to sympathetic dominance over parasympathetic, resulting in a global reduction in HRV and poor prognosis. The underlying mechanism is complex and involves the direct and indirect effects of nicotine and non-nicotine components associated with increased ROS production in central and peripheral areas involved in cardiac sympathetic control. Additionally, the neural–hematopoietic inflammatory axis and the inflammatory reflex further increase OS, establishing a feedback mechanism that perpetuates sympathoexcitation and consequently raises the risk of hypertension, arrhythmias, myocardial infarction, heart failure, and stroke. Therefore, this review highlights the correlation between the HRV indices and OS: indices of cardiac sympathetic modulation correlate directly with plasma OS parameters, while parasympathetic indices correlate inversely. This relationship indicates a high cardiovascular risk in conventional cigarette and e-cigarette smokers (Figure 1). Despite comparatively lower levels of toxicants in ENDSs, their regular use still elicits measurable OS changes, posing significant health risks similar to those observed in conventional cigarette users. Although the available literature is limited, the growing popularity of e-cigs, especially among adolescents and young adults, highlights the urgent need for thorough longitudinal studies. Therefore, the present review reinforces the need for public health policies aimed at reducing smoking exposure and highlights the importance of early autonomic assessment as a potential preventative strategy.

## Figures and Tables

**Figure 1 antioxidants-14-01516-f001:**
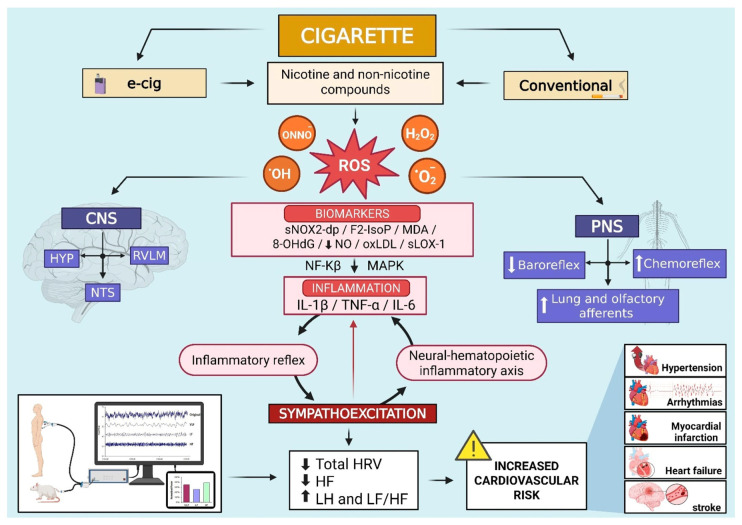
Schematic illustration from both human studies and experimental models suggesting that electronic and conventional cigarettes, by means of nicotine and non-nicotine compounds, increase the production of reactive oxidative species (ROS) in key central and peripheral areas of the nervous system associated with cardiovascular function and basal sympathetic activity. The activation of redox-sensitive signalling (NF-κB and MAPK) facilitates inflammation and stimulates the neural–hematopoietic inflammatory axis and the inflammatory reflex, inducing sympathoexcitation. These processes further increase oxidative stress, establishing a self-sustaining cycle between sympathetic nerve activity and increased ROS production. As a result, HRV decreases, particularly in time domain indices and HF component, while LF values and the LF/HF ratio rise. This pattern is linked to an elevated risk for cardiovascular diseases such as hypertension, arrhythmias, myocardial infarction, heart failure, and stroke. Abbreviations: ^•^OH, hydroxyl radicals; 8-OHdG, 8-hydroxy-2′-deoxyguanosine; CNS, central nervous system; e-cig, electronic cigarette; F2-IsoP, F2-isoprostanes; H_2_O_2_, hydrogen peroxide; HF, high frequency; HRV, heart rate variability; HYP, hypothalamus; IL-1β, interleukin-1beta; IL-6, interleukin-6; LF, low frequency; LF/HF, low frequency/high frequency ratio; MAPK, mitogen-activated protein kinase; MDA, malondialdehyde; NF-Kβ, nuclear factor kappa-light-chain-enhancer of activated B cells; NO, nitric oxide; NTS, nucleus tractus solitarius; O_2_^•−^, superoxide anions; ONNO^−^, peroxynitrite; oxLDL, low-density lipoprotein oxidizability; PNS, peripheral nervous system; ROS, reactive oxygen species; RVLM, rostral ventrolateral medulla; sLOX-1, soluble lectin-like oxidized low-density lipoprotein receptor-1; sNOX2-dp, soluble NOX2-derived peptide; TNF-α, tumour necrosis factor alpha. Symbols: decrease (↓); increase (↑).

**Figure 2 antioxidants-14-01516-f002:**
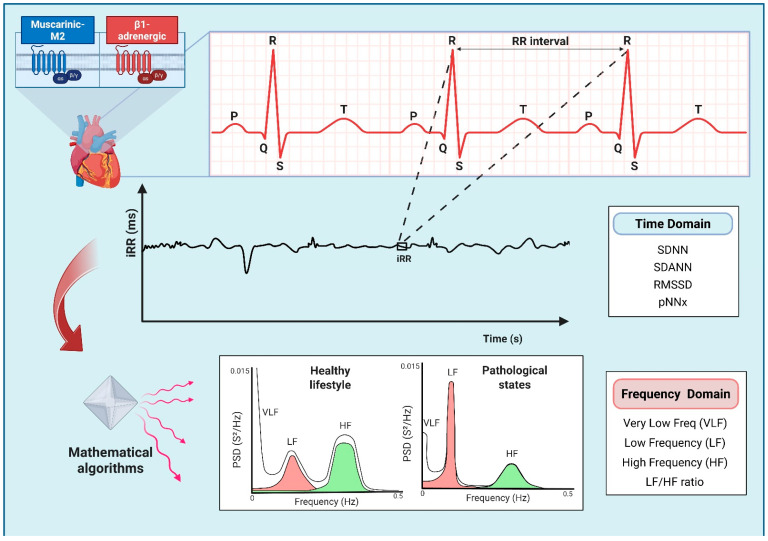
Diagram illustrating the Heart Rate Variability (HRV) analysis used to assess sympathetic and parasympathetic influences on the sinoatrial node by means of muscarinic M2 and β1-adrenergic receptor activation. A 24 h electrocardiogram (ECG) recording allows the extraction of successive R-R intervals (iRRs) and enables comprehensive analysis of the heart rate (HR) variations. The linear method of the HRV analysis includes the evaluation of the time and frequency domain parameters. Time domain indexes measure the HR oscillations over time, while frequency domain indexes analyze the HR frequency distribution using power spectral density (PSD). Mathematical algorithms, e.g., the Fast Fourier Transformation or autoregressive modelling, can assist in visualizing the HRV density distribution. Engaging in a healthy lifestyle is associated with elevated HRV, whereas pathological states may negatively impact it. Abbreviations: pNNx, the percentage of successive NN intervals greater than x ms; RMSSD, root mean square of the successive differences between normal iRRs; SDANNs, standard deviations of the means; SDNN, standard deviation of NN intervals.

## Data Availability

No new data were created or analyzed in this study. Data sharing is not applicable to this article.

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
