# Peer review of "Cardiac Autonomic Dysfunction and Increased Oxidative Stress in Conventional Cigarettes and E-Cigarettes: Heart Rate Variability as a Cardiovascular Predictor"

_antioxidants, 2025, doi:10.3390/antiox14121516_

Round 1
Reviewer 1 Report
The authors present a narrative review addressing the impact of conventional cigarette and e-cigarette consumption on oxidative stress and autonomic heart function, with a particular focus on heart rate variability (HRV). The topic is of potential interest, as it may contribute to understanding redox impairment–driven cardiovascular risks associated with smoking. However, the manuscript in its current form would benefit from improvements in clarity, structure, and depth.
The authors highlight HRV as a sensitive tool to detect oxidative stress–induced cardiac alterations related to smoking. While this is an interesting perspective, the rationale requires further refinement. Sympathetic activation resulting from smoking is not exclusively mediated by oxidative stress but also by other mechanisms, including the direct pharmacological actions of nicotine. Thus, attributing HRV alterations solely to oxidative stress may not fully capture the complexity of the underlying processes. A more balanced discussion that acknowledges these multiple pathways would strengthen the argument.
In its present state, the manuscript does not sufficiently convey conceptual novelty or provide in-depth critical insights. Expanding the discussion to integrate alternative mechanisms and offering clearer organization would enhance its overall contribution to the field.
None
Author Response
Reviewer #1:
Comment: The conclusion contains a significant logical flaw.
The quality of the writing requires substantial improvement.
The conclusion contains a flaw in logic, making it difficult to meaningfully evaluate its validity.
Answer: We thank your observation. We improved the writing and delimitated the conclusion to make it more valid.
Comment: The content is primarily descriptive, making it difficult to identify a sufficiently rigorous and insightful discussion.
The manuscript provides limited novelty, and in-depth insights are not readily apparent.
Answer: We appreciate your comment. As suggested, we integrated in-depth insights into the discussion related to alternative mechanisms involved in the cardiac autonomic dysfunction associated with conventional and electronic cigarettes. We sincerely hope that these improvements sufficiently convey conceptual novelty or provide in-depth critical insights
Comment: The authors present a narrative review addressing the impact of conventional cigarette and e-cigarette consumption on oxidative stress and autonomic heart function, with a particular focus on heart rate variability (HRV). The topic is of potential interest, as it may contribute to understanding redox impairment–driven cardiovascular risks associated with smoking. However, the manuscript in its current form would benefit from improvements in clarity, structure, and depth.
Answer: Thanks for your comment. As requested, we revised the entire manuscript to improve clarity, structure and depth. We rearranged the chapter structure to ensure greater flow to the reader, now presenting: 1. Introduction; 2. Cardiac Autonomic Regulation and Heart Rate Variability; 3. Heart Rate Variability as a health predictor; 4. Impact of conventional and electronic cigarettes on cardiorespiratory systems; 5. Effect of conventional and electronic cigarettes on oxidative stress; 6. The role of oxidative stress in cardiac autonomic dysfunction; 7. Cardiac autonomic dysfunction in conventional and electronic cigarette smokers and the role of oxidative stress. In addition, we added more detail to the presented topics to deepen the revised manuscript.
Comment: The authors highlight HRV as a sensitive tool to detect oxidative stress–induced cardiac alterations related to smoking. While this is an interesting perspective, the rationale requires further refinement. Sympathetic activation resulting from smoking is not exclusively mediated by oxidative stress but also by other mechanisms, including the direct pharmacological actions of nicotine. Thus, attributing HRV alterations solely to oxidative stress may not fully capture the complexity of the underlying processes. A more balanced discussion that acknowledges these multiple pathways would strengthen the argument.
Answer: Thanks for the appropriate suggestion. In the first version of the manuscript, we had already mentioned the main effect of nicotine, but it was not very clear. In the revised version, we rewritten this part of the text to emphasize the direct pharmacological action of nicotine as suggested by the reviewer.
“Clinical data have shown that CCS exposure promotes cardiac autonomic dysfunction characterized by sympathetic overactivation. The main mechanism involved in this autonomic imbalance is the direct pharmacological action of nicotine in its receptors located in peripheral autonomic ganglia and the adrenal medulla, increasing catecholamine release [86,88,98–100]. Sympathoexcitatory effects may also result from the influence of nicotine on chemoreflexes [103]. Argacha et al. [104] found that nicotine enhances peripheral chemoreflex sensitivity to hypoxia in non-smokers, which increases sympathetic drive.
Non-nicotine components may also cause sympathoexcitation by upregulating nicotine receptors [101] or inhibit monoamine oxidase to reduce catecholamine degradation [102]. Additionally, non-nicotine smoke inhalation can activate olfactory afferents which activates both sympathetic and parasympathetic responses, affecting cardiovascular and respiratory functions in conscious rats [84]. Of note, Middlekauff, Park and Moheimani [86] proposed that non-nicotine components, such as fine particulate matter, associated with the direct effect of nicotine, are both responsible for sympathoexcitation in smokers by means of a complex mechanism involving OS, inflammation, decrease in nitric oxide production and baroreflex suppression. These mechanisms become persistent via a positive feedback loop between sympathetic nerve activity and increase in ROS production. Thus, in addition to the effect of nicotine, OS increase is a crucial mechanism which contributes to sympathetic overactivation. This model has significant implications for other emissions, such as e-cig, hookahs, and air pollution [87].” (page 8-9, lines 398-419 in clean version)
“As in tobacco smoking, studies demonstrated that nicotine is the main instigator of the cardiovascular adverse effects that accompany ENDS [109-112]. Nguyen et al. [109] demonstrated that higher levels of nicotine acutely delivered by the 4th generation e-cig increased HR and AP and produced changes in the HRV pattern associated with sympathetic overactivity. Nonetheless, there are data indicating that both nicotine and non-nicotine e-cig components can increase cardiac risk [111]. Nagy et al. [111], by means of a systematic review of case reports, identified cardiovascular complications associated with both nicotine and non-nicotine vaping, with half of the cases resulting in cardiac arrest likely due to arrhythmias from sympathetic predominance. On the other hand, in a systematic review including 19 studies, Garcia, Gornbein, and Middlekauff [110] found that acute vaping increased HR and AP less than CS, and nicotine but not non-nicotine constituents in e-cig aerosol were responsible for the sympathoexcitatory effects (page 9-10, lines 447-458 in clean version)
Comment: In its present state, the manuscript does not sufficiently convey conceptual novelty or provide in-depth critical insights. Expanding the discussion to integrate alternative mechanisms and offering clearer organization would enhance its overall contribution to the field.
Answer: As aforementioned, we revised the entire manuscript to improve clarity, organization and depth. As suggested, we integrated alternative mechanisms involved in the cardiac autonomic dysfunction associated with conventional and electronic cigarettes into the discussion. We sincerely hope that these improvements sufficiently convey conceptual novelty or provide in-depth critical insights
Reviewer 2 Report
The authors aimed to review the literature on the impact of cigarette and electronic nicotine delivery system (ENDS) consumption on autonomic heart dysfunction.
General comment: The text is difficult to read, and some chapters are not clearly linked. It generates confusion for the reader. I suggest revising the entire manuscript to better harmonize and complete the different chapters.
More specifically:
Lines 84-88: While e-cig…intracellular ROS… Something does not seem right. The two phrases contradict each other. Please revise.
Chapter 3 is quite short. I suggest improving it.
Additionally, chapters 3 and 4 are not connected. They seem to belong to different papers. Please try to harmonize them better and introduce the topics accordingly.
Line 130: Define HRV (heart rate variability).
Author Response
Reviewer #2:
Comment: The text is difficult to read, and some chapters are not clearly linked. It generates confusion for the reader. I suggest revising the entire manuscript to better harmonize and complete the different chapters.
Answer: We appreciate your comment. As requested, we revised the entire manuscript, inserted more details in the topics presented and we organized them to become more clearly linked. In the amended manuscript we reorganised the manuscript into the following sections: 1. Introduction; 2. Cardiac Autonomic Regulation and Heart Rate Variability; 3. Heart Rate Variability as a health predictor; 4. Impact of conventional and electronic cigarettes on cardiorespiratory systems; 5. Effect of conventional and electronic cigarettes on oxidative stress; 6. The role of oxidative stress in cardiac autonomic dysfunction; 7. Cardiac autonomic dysfunction in conventional and electronic cigarette smokers and the role of oxidative stress.
Detailed comments: Lines 84-88: While e-cig…intracellular ROS… Something does not seem right. The two phrases contradict each other. Please revise.
Answer: We thank your observation. The paragraphs “While e-cig…intracellular ROS...” was rewritten.
“Some studies demonstrated that although e-cig led to a significant degree of alterations in OS, it seemed to have less pronounced adverse effects as compared with CCS [68]. Moheimani et al. [69,] Shahab et al. [70] and Christensen et al. [63] showed that e-cig are responsible for elevated levels of F2-isoprostane and low-density lipoprotein oxidizability, biomarkers of OS and inflammation, in comparison to nonsmokers [64]. Conversely, Taylor et al. [67] observed that exposure to e-cig aerosol extracts did not result in a significant rise in intracellular ROS or depletion of glutathione in bronchial epithelial cells, unlike CCS, which induced an 83% increase in ROS and a 98.6% reduction in the GSH:GSSG ratio. Nonetheless, subsequent studies confirm that chronic exposure to e-cig vapor can still induce oxidative DNA damage, mitochondrial dysfunction, and overexpression of pro-inflammatory genes [62]..” (page 6, lines 274-285 in clean version)
Detailed comments: Chapter 3 is quite short. I suggest improving it.
Answer: Thanks for the appropriate suggestion. Chapter 3 was improved and additional information was inserted as requested.
Detailed comments: Additionally, chapters 3 and 4 are not connected. They seem to belong to different papers. Please try to harmonize them better and introduce the topics accordingly.
Answer: We thank your observation. In fact, chapters 3 and 4 address different subjects and they are not directly related. Nevertheless, we reorganized the topics and inserted some linking sentences to try to harmonize them.
Detailed comments: Line 130: Define HRV (heart rate variability).
Answer: The HRV definition was inserted in the amended manuscript. “This method is known as the HRV and it is defined as the assessment of oscillations in the successive heartbeats intervals, using the R-R intervals (iRR).” (page 2, lines 71-72 in clean version)
Round 2
Reviewer 2 Report
I appreciate the changes and the effort the authors made to address my comments.
Some minor points are still present:
Please, check punctuation. Some commas are missing here and there, and read the text carefully to correct minor misspellings or imprecisions.
Line 292. …although-cig led… something is missing
Author Response
Reviewer #2:
Comment: Please, check punctuation. Some commas are missing here and there, and read the text carefully to correct minor misspellings or imprecisions.
Answer: We thank your observation. We checked the punctuation and corrected minor spelling errors or inaccuracies. Additionally, an English-language editor reviewed the entire English text.
Comment: Line 292. …although-cig led… something is missing
Answer: We did not find this part in line 292 of either version. We believe that the reviewer is referring to the part of the text in lines 274-276, as follows: “Some studies demonstrated that although e-cig led to a significant degree of alterations in OS, it seemed to have less pronounced adverse effects as compared with CCS [68].”